# Engineering exosomes with iRGD for targeted RNAi therapy against pancreatic cancer mediated by long non-coding RNA PLBD1-AS1

Wenbo Zhu[1]☯, Xintong Zhao[1]☯, Weina Hao[1]☯, Xianzhu Zhou[1], Congjia Ma[1], Jiayu Chen[1], Yating Zhao[1], Xiangyu Kong[1]‡*, Yiqi Du[1]‡*, Lei Li[2]‡*

**1** Department of Gastroenterology, Changhai Hospital, Naval Medical University, Shanghai, China,
**2** Digestive Endoscopy Center, Shanghai Tenth People's Hospital, Tongji University School of Medicine, Shanghai, China

☯ These authors contributed equally to this work.
‡ XK, YD, and LL also contributed equally to this work.
* xiangyukong185@hotmail.com (XK); duyiqi@hotmail.com (YD); raulowen84@hotmail.com (LL)

## Abstract

Tumor-derived exosomes play critical roles in pancreatic ductal adenocarcinoma (PDAC) progression by mediating intercellular communication within the tumor microenvironment. This study identifies the long non-coding RNA PLBD1-AS1 as a functional oncogenic lncRNA enriched in PDAC exosomes. We demonstrate that PLBD1-AS1 promotes tumor cell proliferation, migration, and invasion by interacting with the glycolytic enzyme ALDOA and enhancing glycolytic flux. Furthermore, tumor exosomes deliver PLBD1-AS1 to pancreatic stellate cells (PSC), augmenting their glycolysis and facilitating their activation into cancer-associated fibroblasts, thereby shaping a pro-tumorigenic microenvironment. To target it, we developed an engineered exosome system modified with the tumor-penetrating peptide iRGD for specific delivery of siPLBD1-AS1 to both tumor and stromal cells. The resulting iRGD-exo-siPLBD1-AS1 construct demonstrated enhanced cellular uptake and effectively suppressed PLBD1-AS1 expression, inhibited glycolysis, impaired PSC activation, and significantly attenuated tumor growth. Our findings reveal a novel mechanism of exosome-mediated metabolic crosstalk in PDAC and establish a promising RNAi-based therapeutic strategy targeting this lethal malignancy.

## Introduction

Pancreatic ductal adenocarcinoma (PDAC) is a malignancy with an exceptionally poor prognosis, as evidenced by a 5-year overall survival rate of approximately 10% [1]. A hallmark of PDAC is its extensive stromal desmoplasia, comprising various stromal components such as cancer-associated fibroblasts (CAFs), immune cells, and endothelial cells. These stromal components interact with tumor cells, thereby establishing a tumor microenvironment that promotes tumor progression [2].

**Data availability statement:** All relevant data are within the paper and its Supporting information files.

**Funding:** Supported in part by grants from the National Natural Science Foundation of China (82172572 to L.L., 82170659 to Y.D., 82072760 and 82473057 to X.K.). These grants provided essential financial support for the conduct of this research, including personnel support, experimental materials, and data analysis.

**Competing interests:** The authors have declared that no competing interests exist.

Activated pancreatic stellate cells (PSCs) are pivotal in generating the extracellular matrix in PDAC and constitute the primary source of CAFs [3]. Tumor cells can induce the activation of PSCs into CAFs, thereby accelerating disease progression [4,5]. The crosstalk between PSCs and tumor cells establishes a vicious cycle that further exacerbates malignancy [6,7]. Consequently, elucidating the mechanisms driving PSC activation may reveal promising therapeutic strategies to curb tumor progression.

Recent studies have demonstrated that tumor cells can influence neighboring cells not only through secretory soluble protein but also via exosomes that carry diverse molecular cargoes, including proteins and nucleic acids [8–10]. Tumor-derived exosomes are typically more abundant than those from normal cells [11,12], and can be internalized by recipient cells to promote tumorigenesis and progression [13,14]. Building on this, we sought to identify functional exosomal RNAs that drive tumor progression and to exploit engineered exosomes for RNA interference (RNAi)-based therapy. Exosomes have emerged as excellent vectors for therapeutic applications. Their natural lipid bilayer protects encapsulated nucleic acids from degradation, enabling efficient in vivo delivery [15]. Furthermore, surface engineering can enhance their tissue targeting specificity [16,17], allowing them to reach otherwise challenging sites such as the central nervous system [18] or the core of solid tumors [19], while minimizing uptake by non-target organs [20].

Here we report that the long non-coding RNA PLBD1-AS1 is linked to poor prognosis in PDAC. Mechanistically, PLBD1-AS1 is delivered via tumor-derived exosomes, where it promotes malignant properties in tumor cells and drives PSC activation into CAFs by augmenting cellular glycolysis. To exploit its therapeutic potential, we developed exosomes functionalized with the tumor-penetrating peptide iRGD (cyclic CRGDKGPDC) to deliver siPLBD1-AS1 specifically to tumor cells and PSCs. The iRGD peptide selectively binds integrin αvβ3 and neuropilin-1 (NRP-1) [21], enabling tumor-specific delivery and enhancing penetration into the dense PDAC stroma. Overall, our finding provides a promising RNAi-based strategy for the treatment of pancreatic cancer.

## Materials and methods

### Collection and preprocessing of dataset

The exosomal RNA-seq dataset was obtained from the Gene Expression Omnibus (GEO) under accession number GSE175436. This dataset includes exosomal lncRNA and mRNA expression profiles derived from the plasma of 5 patients with pancreatic ductal adenocarcinoma (PDAC) and 5 healthy donors. For survival and prognostic analysis, clinical and transcriptomic data from the TCGA-PAAD cohort were downloaded via the Xena platform. Only patients with complete survival information and RNA-seq data were included in the analysis. The RNA-seq values from TCGA were provided in Transcripts Per Million (TPM) format.

### Identification of differentially expressed exosomal genes in PDAC

Differentially expressed genes (DEGs) between PDAC and normal samples from GSE175436 were identified using the limma package (v 3.50.1) [22]. The p-values

were adjusted for multiple testing using the Benjamini-Hochberg false discovery rate (FDR) method, and DEGs were defined with the following thresholds: adjusted p (FDR) < 0.05 and |log2Fold Change (FC)| > 1. A total of 2,266 DEGs were obtained. Univariate Cox regression analysis was first applied to screen for genes significantly associated with patient progression, from which 12 candidate genes were selected. These genes were further refined using least absolute shrinkage and selection operator (LASSO) logistic regression with 10-fold cross-validation, implemented via the R glmnet package [23].

## Statistical analysis

All quantitative experiments were conducted with at least three independent biological replicates, and data are presented as the mean ± standard deviation (SD) or standard error of the mean (SEM). Randomization was applied to animal grouping, the order of sample processing in vitro, and the selection of microscopic fields to minimize systematic bias.

Statistical analyses were performed using both R software (version 4.5.2) and GraphPad Prism (version 10.4.2). Differences in survival among patient cohorts were evaluated by the Kaplan-Meier method and compared using the log-rank test. The prognostic value of individual variables was assessed by univariate Cox proportional hazards regression analysis. Correlation analyses were conducted using either Pearson's (for normally distributed data) or Spearman's (for non-parametric data) correlation coefficients.

For comparisons between two groups, the unpaired two-tailed Student's *t*-test (for parametric data) or the Mann-Whitney U test (for non-parametric data) was used. Comparisons among multiple groups were performed using one-way ANOVA (followed by an appropriate post-hoc test such as Tukey's) for parametric variables or the Kruskal-Wallis test for non-parametric variables. Specific statistical tests, the exact number of replicates (*n*), and post-hoc tests used are detailed in the respective figure legends. A two-sided *p*-value of less than 0.05 was considered statistically significant.

## Cell culture

MIA PaCa-2, PANC-1, HPEN and PSCs were obtained from the American Type Culture Collection. Cultured cells were maintained at 37°C in a 5% CO2 humidified atmosphere and grown in DMEM with 10% FBS, 100 units/mL penicillin, and 100 mg/mL streptomycin. To remove exogenous bovine exosomes, FBS was filtered through a 0.22 μm filter, and the filtrate was subjected to ultracentrifugation at 120,000 × g for 90 min at 4°C. The resulting supernatant was collected as exosome-depleted FBS.

## Isolation of exosomes from the medium

Cell culture medium was differentially centrifuged to isolate exosomes from cells. The cell culture was centrifuged at 500 × g for 5 minutes and 2000 × g for 30 minutes to remove the cells, followed by centrifugation at 100,000 × g for 60 minutes and 120,000 × g for 70 minutes, all at 4°C. Exosomes were resuspended in 1 × PBS and stored at 4°C.

## Engineered exosomes

Based on the principle and method of lipid anchoring [24–26], the exosomes were engineered with iRGD peptides using a commercial kit (Cat# EA-08–1, Echobiotech, China). Briefly, 1.5 mL of exosome suspension ($1 \times 10^{10}$ particles/mL) was mixed with 100 μL of an iRGD-PEG-DSPE (Cat# 320701, meloPEG, China) solution in the provided reaction buffer. The mixture was incubated at 25 °C for 3 hours, followed by a subsequent incubation at 4 °C for 24 hours. To remove unreacted components, the sample was purified using a 100 kDa molecular weight cut-off ultrafiltration tube. This involved resuspension in 3.5 mL of washing buffer and centrifugation at 4,000 g for 30 minutes at room temperature. Finally, the purified iRGD-modified exosomes (iRGD-exo) were collected from the filter membrane by gentle pipetting and transferred to a fresh microcentrifuge tube.

siRNA was loaded into exosomes by using an exo-load kit (Cat# ELSR-06, Echobiotech, China). The loading of siRNA into exosomes was performed using an ETP-based incubation method. Briefly, 100 μL of exosomes (1 mg/mL) were mixed with 10 nmol of siRNA and 200 μL of ETP reagent, followed by incubation at 37 °C for 2 hours. The mixture was then concentrated using a 100 kDa molecular weight cut-off ultrafiltration device at 4,000 × g until the volume was reduced to approximately 100 μL. The resulting concentrate was washed twice with washing buffer to remove unencapsulated siRNA and other non-incorporated components. The final purified, siRNA-loaded exosomes were collected for subsequent PLBD1-AS1 analysis and functional studies.

### ELISA test of IL-6, MCP-1, FN

Human IL-6 (Cat# ab178013, Abcam, USA), MCP-1 (Cat# ab179886, Abcam, USA) and FN (Cat# ab219046, Abcam, USA) ELISA kits were obtained from Abcam. Human IL-6, MCP-1 and FN concentrations in the cultured cell conditioned medium were measured in triplicate using ELISA, following the manufacturer's protocol.

### EDU proliferation assay

As directed by the manufacturer, an EdU proliferation assay was conducted using the Cell-Light EdU Apollo 567 *In Vitro* Imaging Kit (Cat# C10310-1, RiboBio, China). The indicated number of cells were seeded into 96-well plates followed by the addition of 100 μL of medium containing 50μM EdU to each well, after which the cells were incubated for 2 hours at 37°C with the medium that contained 50μM EdU. 4% paraformaldehyde was used for fixation and a Hoechst and Apollo reaction cocktail for staining. Images were captured using a fluorescence microscope. The ratio of EdU-positive cells to total cells in each field was calculated.

### Exosome and cell labeling

The fluorescent dyes DiL (Cat# C1036, Beyotime, China) were employed to label exosomes. Exosomes were labeled with DiL for 15 minutes and centrifuged at 100,000 × g for 30 minutes to eliminate free dye. The labeled exosomes were washed twice with PBS at 100,000 × g for 30 minutes each, and subsequently resuspended in PBS.

The fluorescent dyes Phalloidin-FITC (Cat# C5782, Biotechne, China) and DAPI (Cat# C1005, Beyotime, China) were used to label cells. Cells were labeled with Phalloidin-FITC and DAPI for 15 minutes. The labeled cells were washed twice with PBS for 15 minutes to remove any free dye.

### Cell proliferation and colony formation assay

A CCK-8 kit (Cat# C0037, Beyotime, China) was utilized. Cells and exosomes were seeded into 96-well plates with various pretreatments. Subsequently, varying oxaliplatin concentrations were introduced. Following a 48-hour incubation, 10 μL of CCK-8 reagent was added and incubated at 37°C for 2–3 hours. Optical density (OD) values at 460 nm were then measured using a Thermo microplate reader. For the colony formation assay, 300 PANC-1 and MIA PaCa-2 cells with exosomes were seeded into 60 mm dishes. The cells were incubated at 37°C with 5% CO2/95% air for 14 days. Methanol was applied to fix the colonies for 15 minutes. After three PBS washes, colonies were stained with 0.1% crystal violet for 15 minutes at room temperature. Colonies were counted using a light microscope.

### Tumor-cell migration/invasion assay

Cell scratch-wound (horizontal migration) and transwell (vertical invasion) assays were conducted to assess the migratory and invasive capabilities of MIA PaCa2 and PANC-1. The scratch-wound tests were carried out by culturing exosome-treated cells in six-well plates and creating a wound with a 10-μL pipette tip on the monolayer. Subsequently, the cells were incubated for 12–36 hours. Cell migration in the wounded monolayer was evaluated by photographing the cells

at various time points and measuring gap sizes across multiple fields. The transwell assay utilized 24-well tissue culture plates with 12 cell culture inserts (Millipore). A thin layer of basement membrane matrix was pre-coated on each insert, covering the 8-μm-pore-size polycarbonate membrane. In the lower chambers, 10% fetal bovine serum was used to chemoattract cells. The upper chambers were filled with $5 \times 10^5$ cells and exosomes (100 μg) in 300 μL of serum-free medium, and they were incubated at 37°C for 48 hours. Under a microscope, cells that invaded the ECMatrix and migrated through the polycarbonate membrane were stained, counted, and photographed.

## IHC and H&E staining

IHC(Immunohistochemistry) staining was performed by incubating slides overnight with anti-Ki67(1:5000, Cat# 28074–1-AP, Proteintech, China) and αSMA (1:1000, Cat# 14395–1-AP, Proteintech, China) antibodies at 4°C followed by 1 hour at room temperature with a secondary antibody conjugated to rabbit HRP. In the final step, positive staining was visualized using DAB substrate liquid, followed by hematoxylin counterstaining. For histological examination, hematoxylin and eosin (H&E) staining was performed on liver and spleen sections from mice. Briefly, deparaffinized and rehydrated tissue sections were stained with hematoxylin to visualize nuclei, followed by eosin counterstaining to highlight cytoplasmic and extracellular structures.

## RNA pull-down and mass spectrometry

To identify proteins that interact with PLBD1-AS1, RNA pull-down assays followed by mass spectrometry (MS) were performed. Biotinylated PLBD1-AS1 RNA for the pull-down assay was generated by in vitro transcription using the MAX-Iscript® T7 Kit (Cat# AM1312, Thermo Fisher Scientific, USA). According to the manufacturer's protocol, 1 μg of linearized plasmid template was transcribed in a 20 μL reaction containing 0.5 mM of ATP, CTP, GTP, a mixture of 0.5 mM UTP/biotin-16-UTP, and T7 RNA polymerase for 1 h at 37°C. After DNase I treatment to remove the template, the RNA probe was purified from unincorporated nucleotides using NucAway™ Size-Exclusion Spin Columns (Cat# AM10070, Thermo Fisher Scientific, USA).

For the RNA pull-down assay, total protein was extracted from PANC-1 and MIA PaCa-2 cells using IP lysis buffer containing a protease inhibitor cocktail. A total of 500 μg of protein lysate per sample was incubated with 2 μg of purified biotinylated RNA for 1 hour at room temperature to facilitate RNA-protein complex formation. Streptavidin magnetic beads (Cat# 20164, Thermo Fisher Scientific, USA) from the Pierce™ Magnetic RNA-Protein Pull-Down Kit were then added to capture the complexes, followed by a 30-minute incubation. The beads were subsequently washed extensively with wash buffer to remove nonspecifically bound proteins.

The bound proteins were eluted and prepared for two downstream analyses: (1) A portion of the eluate was separated by SDS-PAGE and visualized by silver staining for a qualitative assessment of pulled-down proteins. (2) The remaining eluate was subjected to analysis by liquid chromatography with tandem mass spectrometry (LC-MS/MS) to identify the specific interacting proteins. Proteins enriched in the PLBD1-AS1 pull-down compared to the antisense control group were considered potential binding partners. For MS analysis, samples were digested into peptides, which were then separated on an EASY-nLC 1200 UHPLC system using an increasing gradient of acetonitrile. The eluted peptides were directly ionized and analyzed by a Q Exactive™ HF-X mass spectrometer. The generated MS/MS data were searched against a human database using MaxQuant (v1.6.15.0) with standard settings and an FDR threshold of 1%. Putative PLBD1-AS1 binding partners were identified by significant enrichment over the antisense control.

## RIP-qPCR

RIP assay was performed using the PureBinding® RNA Immunoprecipitation Kit (Cat# P0101, GENESEED, China) following the manufacturer's protocol. Antibodies against ALDOA (Cat# ab245469, Abcam, China) and control IgG

(Millipore, USA) were employed. Briefly, protein A/G magnetic beads were first conjugated with either anti-SLU7 antibody or control IgG, and then incubated with cell lysates. Co-precipitated RNA was subsequently isolated and quantified by RT-qPCR.

## Subcutaneous pancreatic cancer transplant model and exosome therapy

Six-week-old female athymic nude BALB/c mice were kept in a pathogen-free environment. The Animal Research Ethics Committee at Naval Medical University (Shanghai, China) preapproved the animal study. The mice were randomly assigned to two groups, with four mice in each group. MIA PaCa-2 and PSCs were injected subcutaneously into mice. Tumor growth was monitored daily. Two weeks later, a mouse of each group was taken photographs after tail vein injection of exosomes stained by DiD by an *in vivo* imaging system (Tanon, Tanon ABL X6). The remaining three mice in each group received 100 micrograms of exosomes/iRGD-exo via tail vein injection every 2 days for 2 weeks. The mice were euthanized after 2 weeks. The exosome dosage was selected based on established protocols from recent preclinical studies [27–29].

## Western blot analysis

Standard Western blotting was performed using whole-cell lysates. The following primary antibodies were used: CD63 (Cat# ab134045, Abcam, China), CD81 (Cat# ab79559, Abcam, China), CD9 (Cat# ab236630, Abcam, China), Calnexin (Cat# ab22595, Abcam, China), AMPK (Cat# ab32047, Abcam, China), p-AMPK (Cat# ab133448, Abcam, China), ALDOA (Cat# ab245469, Abcam, China), ATP6V1B2 (Cat# ab200839, Abcam, China), α-SMA (Cat# ab108424, Abcam, China), VIM (Cat# ab20346, Abcam, China), FAP (Cat# ab314456, Abcam, China), and GAPDH (Cat# ab8245, Abcam, China). Goat anti-rabbit (Cat# ab6721, Abcam, China) or goat anti-mouse (Cat# ab6789, Abcam, China) secondary antibodies were used for detection.

## Transient transfection and RNA interference

To knock down PLBD1-AS1, siRNA specifically targeting PLBD1-AS1 was synthesized and transfected into PDAC cells. Lipofectamine 2000 was used for cell transfection. The cells were cultured for 48 hours after transfection before being used for experiments.

## Total RNA extraction and quantitative RT-PCR

Total RNA was extracted from tissues and cultured cells using TRIzol® reagent (Cat# 15596026, Invitrogen, USA) following the manufacturer's protocol. RNA concentration and purity were measured by UV spectrophotometry, and samples with an A260/A280 ratio between 1.8 and 2.0 were used for further analysis. cDNA was synthesized using the HiScript® III 1st Strand cDNA Synthesis Kit (Cat# R312-02, Vazyme, China). Quantitative PCR was performed using SYBR Green master mix (Cat# Q712-02, Vazyme, China) on a LightCycler 480II system (Roche, Germany). The PCR protocol included initial denaturation at 95°C for 5 min, followed by 40 cycles of 95°C for 10 s and 60°C for 30 s. mRNA and lncRNA expression levels were normalized to β-actin. For exosomal PLBD1-AS1 detection, cel-miR-39 (Cat# AM17100, Thermo Fisher Scientific, USA) was added as a spike-in control during RNA extraction.

 Primer Sequence are followed. Human PLBD1-AS1 Forward: GCCGCCTCTCCGAGTTTT, Human PLBD1-AS1 Reverse: TCCCAGAAATGCCACCTGTC; Human NRP-1 Forward: GGCGCTTTTCGCAACGATAAA, Human NRP-1 Reverse: TCG-CATTTTTCACTTGGGTGAT; Human αvβ3 Forward: ATCTGTGAGGTCGAAACAGGA, Human αvβ3 Reverse: TGGAG-CATACTCAACAGTCTTTG; Human β-ACTIN Forward: AGAGCTACGAGCTGCCTGAC,  Human β-ACTIN Reverse: AGCACTGTGTTGGCGTACAG.

## Transmission electron microscopy (TEM)

20µL of exosomes were added to the copper net, which was then dried in the dark for five to ten minutes before being absorbed by the edge liquid. Subsequently, 20 µL of 2% phosphotungstic acid solution was added and incubated for 3–5 minutes. Exosomes were examined using a HITACHI HT7700 transmission electron microscope.

## Nanoparticle tracking analysis

Exosome sizes and quantities were measured using the NanoSight NS300 system (Malvern, UK). After resuspension in 1X PBS, the exosomes were filtered with a 0.22µm screen. Data analysis was conducted using Nanoparticle Tracking Analysis (NTA) software.

## Measurements of aldolase activity, F1, 6 BP and lactate

Aldolase activity was quantified using an Aldolase Activity Colorimetric Assay Kit (Cat# ab196994, Abcam, China) according to the manufacturer's protocol. Cellular Fructose-1,6-Bisphosphate (F1,6 BP) levels were measured with a Fructose-1,6-Bisphosphate Assay Kit (Cat# ab284537, Abcam, China). Lactate levels were determined using a commercial Lactate Assay Kit (Cat# S01955, Beyotime Biotechnology, China), following the supplied instructions.

## Seahorse assay

Cells were seeded in an XF24 cell culture plate to reach 90% confluence. Prior to the assay, the culture medium was replaced with assay medium and cells were incubated for 1 h at 37 °C. Following baseline measurement, metabolic modulators prepared in assay medium were sequentially injected into each well. The Glycolytic Rate Assay (Cat# 103344−100, Agilent Technologies, USA) and the Cell Mito Stress Test Kit (Cat# 103015−100, Agilent Technologies, USA) were used to measure the extracellular acidification rate (ECAR) and oxygen consumption rate (OCR), respectively, in real time using a Seahorse XFe24 Analyzer (Agilent Technologies). After the assay, cells were fixed and stained with Hoechst 33342, and nuclei were counted using ImageJ for normalization. All OCR and ECAR values are expressed as pmol/min/$10^3$ cells and mpH/min/$10^3$ cells, respectively.

## *In vivo* experiments

All animal experiments involved in this study followed the Arrive guidelines. Female BALB/c nude mice (6−8weeks old, 18-22g) were purchased from GemPharmatech Co., Ltd (Jiangsu, China) and housed under specific pathogen-free (SPF) conditions in individually ventilated cages (22±2°C, 50±10% humidity, 12h light/dark cycle) with autoclaved food and acidified water. All procedures were approved by the Institutional Animal Ethics Committee (CHEC (A.E)2025−008).

Experiment 1 (shPLBD1-AS1 knockdown): Group 1 (Control): MIA PaCa-2 cells transfected with shNC; Group 2 (Experimental): MIA PaCa-2 cells transfected with shPLBD1-AS1 (n = 3/group). Mice were anesthetized with 2% isoflurane, and $5 \times 10^5$ cells in 100 µL PBS were injected subcutaneously into the right flank. Tumor volume was measured every 3 days using digital calipers and calculated as (L × W²)/2.

Experiment 2 (iRGD-Exosome biodistribution): Group 1 (Control): Unmodified exosomes; Group 2 (Experimental): iRGD peptide-conjugated exosomes (n = 3/group). When tumors reached 100–150 mm³, 100 µg DiR-labeled exosomes in 200 µL PBS were injected via tail vein. Biodistribution was monitored at 2, 6, 12, 24, and 48h post-injection using IVIS Spectrum imaging system.

Experiment 3 (Therapeutic efficacy): Group 1 (Control): iRGD-Exosome loaded with scramble siRNA; Group 2 (Experimental): iRGD-Exosome loaded with siPLBD1-AS1 (n = 3/group). Treatment (10 µg exosomes in 200 µL PBS per dose) was administered twice weekly via tail vein injection when tumors reached 50–100 mm³.

The humane endpoints were defined as: 1) tumor volume exceeding 1500 mm³, 2) ulceration or necrosis >20% of tumor surface area, or 3) body weight loss >20% of initial weight. Euthanasia was performed by $CO_2$ asphyxiation followed by cervical dislocation for confirmation.

Data are presented as mean±SEM. Statistical significance was determined by two-way ANOVA with Tukey's post-hoc test using GraphPad Prism 9.0 (GraphPad Software). A p-value <0.05 was considered statistically significant.

## Results and discussion

### Exosomal lncRNA PLBD1-AS1 is associated with poor prognosis in pancreatic cancer

Exosome sequencing data from five healthy individuals and five patients with pancreatic cancer identified 2,266 differentially expressed genes (DEGs), with 1,110 upregulated and 1,156 downregulated (Fig 1A). These DEGs indicate differences in the exosomal RNA composition between pancreatic cancer patients and healthy individuals, suggesting that exosome-mediated intercellular communication may play an important role in the development and progression of pancreatic cancer. Subsequently, to identify prognosis-associated differential exosome genes, we integrated lncRNA and mRNA data from the TCGA database for additional analysis. We integrated the differential exosome genes with their prognostic relevance from the TCGA database. After confirming the proportional hazards assumption, we performed univariate Cox analysis followed by LASSO regression to identify prognosis-associated genes (Figs 1B and  1C). Twelve genes were identified as significantly associated with prognosis. To identify exosome-derived genes from tumor cells that promote cancer progression, we focused on three upregulated candidates: MIR4435−2HG, PLBD1-AS1, and LINC00707. Subsequent TCGA-based prognostic and Cox regression analyses revealed that among these, PLBD1-AS1 demonstrated the most significant correlation with poor patient outcomes (Figs 1D–1F). Finally, we validated the expression of PLBD1-AS1 in pancreatic ductal cells (HPNE) and tumor cells (MIA PaCa-2 and PANC-1), confirming its significant upregulation in tumor cells (Fig 1G).

### PLBD1-AS1 facilitates the proliferation, migration, and invasion of tumor cells

To investigate the impact of PLBD1-AS1 on tumor cells, we employed siRNA to knockdown PLBD1-AS1 in MIA Paca-2 and PANC-1 cell lines. The knockdown of PLBD1-AS1 significantly suppressed tumor cell proliferation and colony-forming ability (Figs 2A-2C). Transwell assays further demonstrated that PLBD1-AS1 knockdown significantly impaired the migratory and invasive capacities of both cell lines (Fig 2D). To extend these findings *in vivo*, we established a subcutaneous xenograft model in mice. Tumors derived from PLBD1-AS1-silenced cells exhibited significantly reduced proliferative activity (Fig 2E).

### PLBD1-AS1 enhances enzymatic activity and promotes glycolysis by interacting with ALODA

To elucidate the mechanism through which PLBD1-AS1 regulates tumor cells, we employed RNA pull-down assays combined with mass spectrometry to identify proteins interacting with PLBD1-AS1. We found that PLBD1-AS1 directly binds to ALDOA (Fig 3A), an enzyme involved in glycolysis and known to facilitate tumor progression [30,31]. The interaction between PLBD1-AS1 and ALDOA was further validated in MIA PaCa-2 and PANC-1 cell lines by immunoblotting and RIP-qPCR (Fig 3B).

Subsequently, we confirmed the association between ALDOA expression and unfavorable tumor prognosis in TCGA-PAAD, suggesting that ALDOA may facilitate tumor progression (S1A Fig). However, ALDOA protein levels remained unchanged upon PLBD1-AS1 knockdown in the MIA Paca-2 and PANC-1 cell lines (Fig 3C and S1B Fig). We therefore hypothesized that PLBD1-AS1 influences ALDOA enzymatic activity. Silencing PLBD1-AS1 in tumor cells led to reduced lactate levels, aldolase activity, and F1,6 BP levels (Figs 3D-F), indicating impaired glycolytic flux. Consistent with this, ECAR and OCR measurements confirmed that PLBD1-AS1 enhances both glycolysis and oxidative phosphorylation in tumor cells (Fig 3G and S1C Fig).

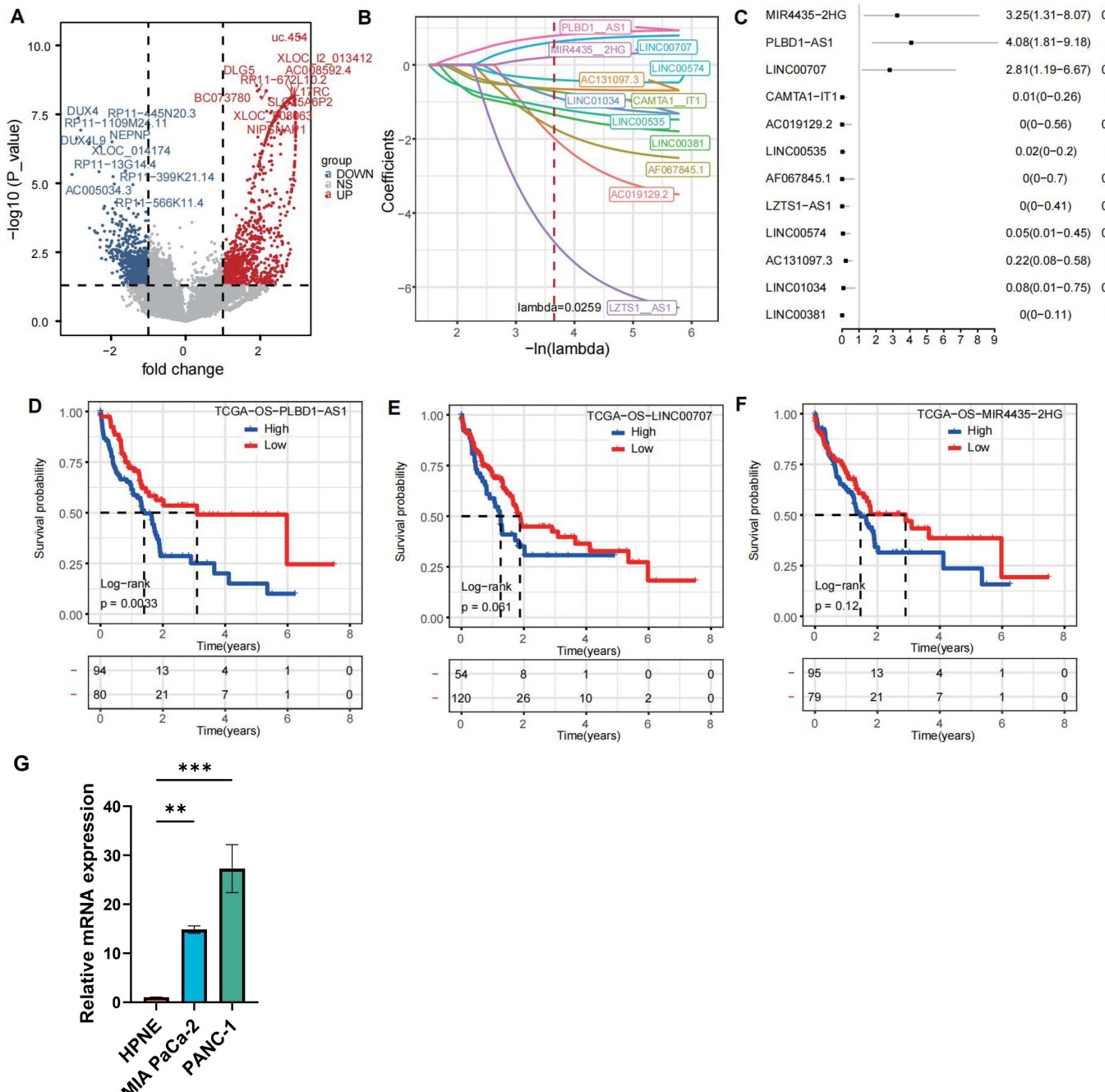

**Fig 1. The extracellular vesicle RNA PLBD1-AS1 is associated with poor prognosis in pancreatic cancer. (A)** Volcano plot of the differentially expressed lncRNAs from exosome RNA-seq data of five healthy donors and five pancreatic cancer patients. **(B)** Cross-sectional profile of LASSO regression coefficients from the TCGA database based on the pool of exosomal DEGs. **(C)** Forest plot of univariate Cox regression analysis for prognostic genes. **(D-F)** Kaplan-Meier survival analysis for PLBD1-AS1, LINC00707, and MIR4435−2HG. **(G)** Relative mRNA expression levels of PLBD-AS1 in HPNE, MIA Paca-2, and Panc-1 cell lines.

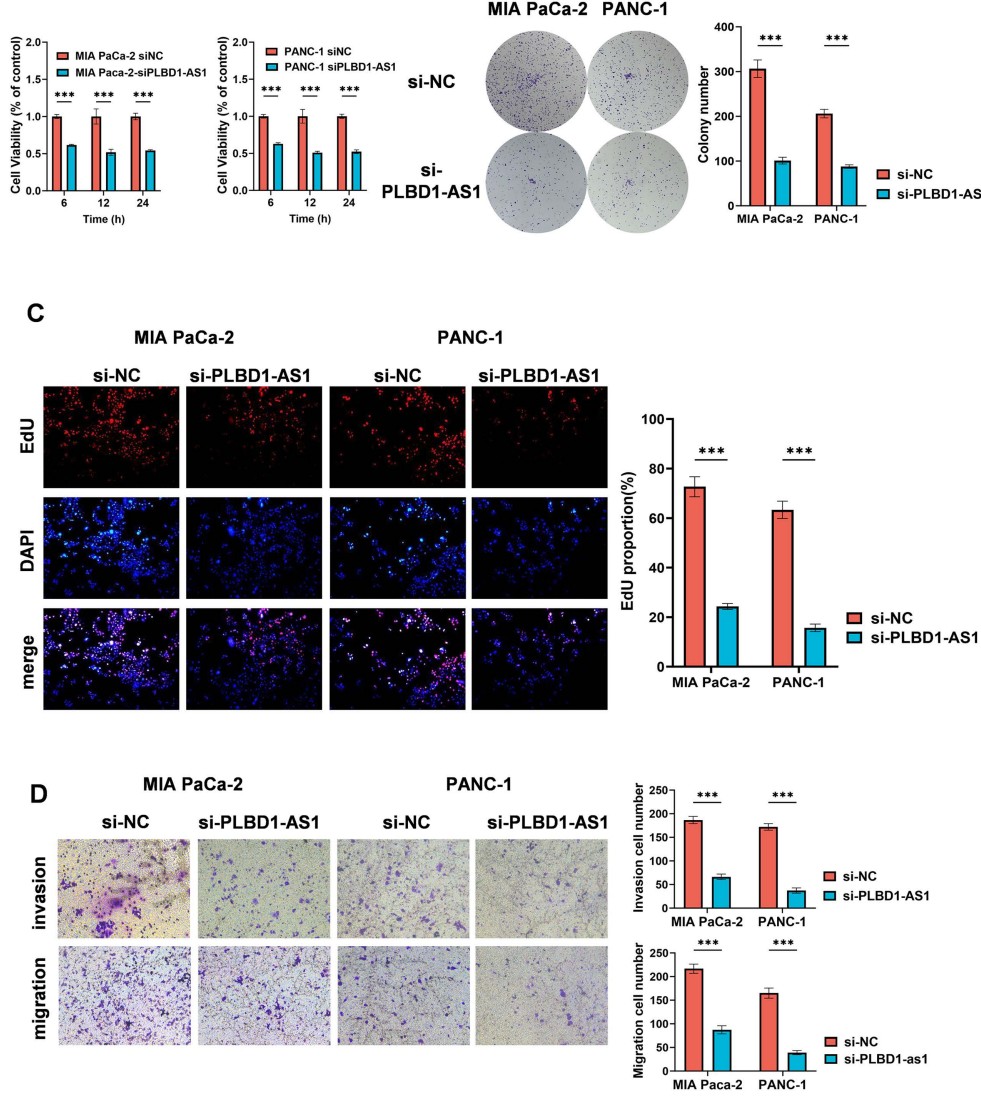

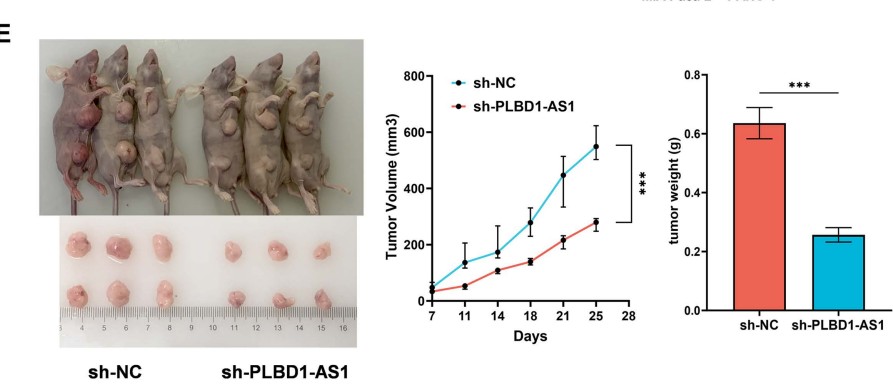

**Fig 2. PLBD1-AS1 promotes tumor cell proliferation, migration, and invasion. (A)** Cell viability of MIA PaCa-2 and PANC-1 cells measured by CCK-8 assay (n = 3). **(B and C)** Representative images and quantification of colony formation and EdU assays in MIA PaCa-2 and PANC-1 cell lines (n = 3). **(D)** Transwell migration and invasion assays in control and PLBD1-AS1-silenced cells (n = 3). **(E)** *In vivo* tumor growth in a subcutaneous xeno-graft model established in BALB/c nude mice using MIA PaCa-2 cells transfected with siPLBD1-AS1 or siNC (n = 3). Statistical analyses were performed by Student's t test. Bar graphs represent mean ± SEM. ns, not significant. *p < 0.05, **p < 0.01, ***p < 0.001.

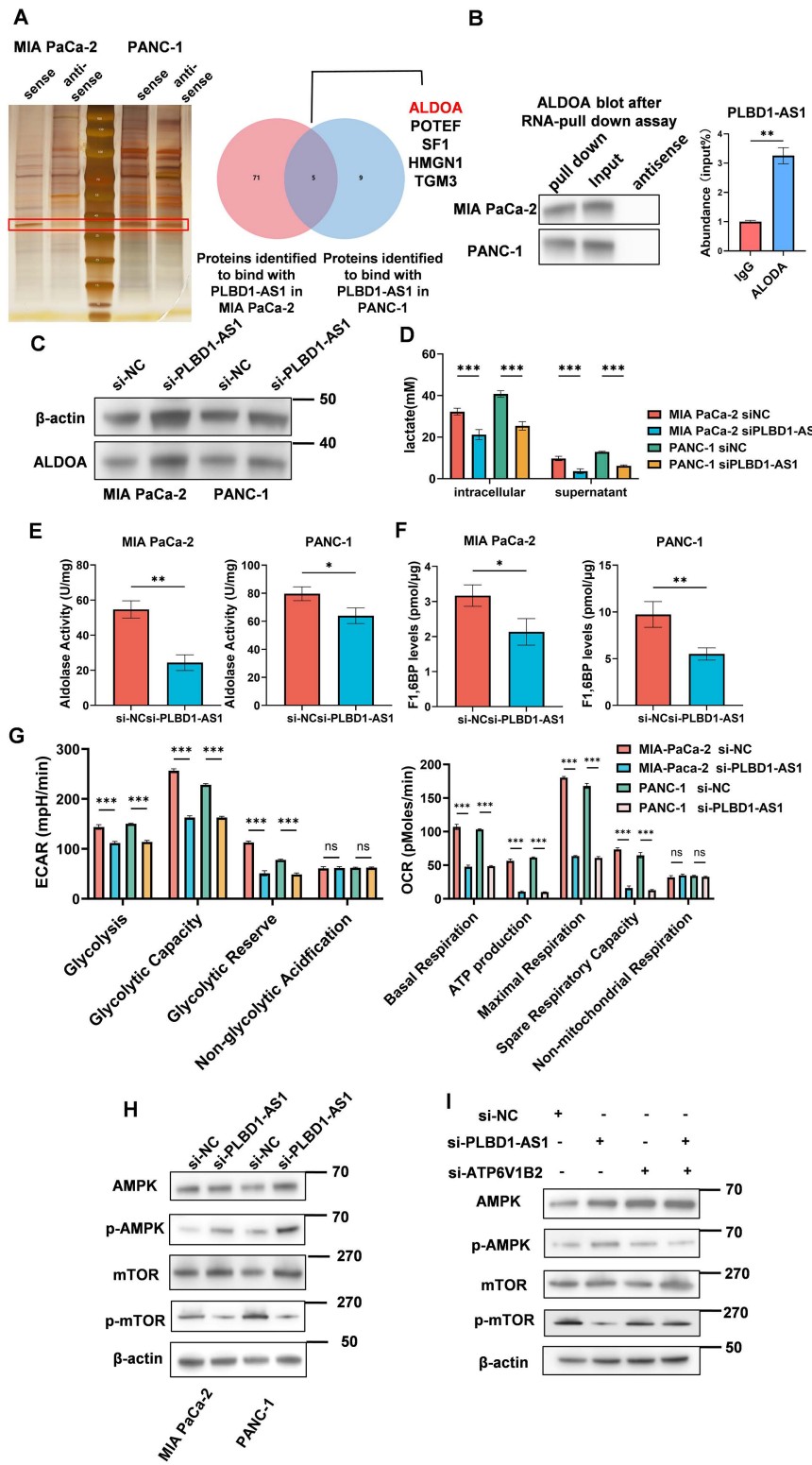

**Fig 3. PLBD1-AS1 interacts with ALDOA to enhance its enzymatic activity and promote glycolysis. (A)** Silver-stained PAGE gel of proteins pulled down with biotin-labeled PLBD1-AS1 in MIA PaCa-2 and PANC-1. **(B)** (left) Western blot confirming the interaction between PLBD1-AS1 and ALDOA, with antisense PLBD1-AS1 as a negative control (n = 3); (right) RIP-qPCR analysis was performed to examine the association between PLBD1-AS1

lncRNA and ALDOA protein. **(C)** Western blot for ALDOA and β-actin in MIA PaCa-2 and PANC-1 cell lines (n = 3). **(D)** Intra- and extracellular lactate levels upon PLBD1-AS1 silencing (n = 3). **(E and F)** ALDOA activity and F-1,6-BP levels after PLBD1-AS1 knockdown (n = 3). **(G)** ECAR and OCR profiles in PLBD1-AS1-silenced cells (n = 3). **(H)** Western blot analysis of AMPK pathway protein phosphorylation following PLBD1-AS1 knockdown (n = 3). **(I)** Western blot of phosphorylated AMPK pathway proteins in cells with single or dual knockdown of PLBD1-AS1 and ATP6V1B2 (n = 3). Statistical analyses were performed by Student's t test (D, E, F and **G**). Bar graphs represent mean ± SEM. ns, not significant. *p < 0.05, **p < 0.01, ***p < 0.001.

Given prior evidence that ALDOA regulates AMPK via interaction with v-ATPase [32], we assessed AMPK signaling upon PLBD1-AS1 silencing. The results indicated that silencing PLBD1-AS1 could promote the activation of AMPK (Fig 3H and S1D Fig). The activation of AMPK necessitates the prior binding of ALDOA and vATPase [33,34]. Therefore, we aimed to investigate if PLBD1-AS1 influenced ALDOA's binding to vATPase. Silencing ATP6V1B2 abolished the AMPK activation induced by PLBD1-AS1 knockdown (Fig 3I and S1E Fig), confirming that PLBD1-AS1 modulates AMPK signaling through ALDOA–v-ATPase.

## PLBD1-AS1-exo promotes PSCs activation towards CAFs

The activation of PSCs towards CAFs is essential for pancreatic cancer progression [35]. To investigate whether PLBD1-AS1-exosomes derived from tumor cells are internalized by PSCs and promote their activation into CAFs, we co-cultured PSCs with DiL-labeled exosomes isolated from MIA PaCa-2 and PANC-1 cells. The results confirmed exosome uptake by PSCs (Fig 4A). Simultaneously, we measured the expression of PLBD1-AS1 in PSCs before and after co-culture with tumor cells and MSCs (S2A Fig). Although PLBD1-AS1 expression was initially low in PSCs, its expression significantly increased after co-culture with tumor cells. Knockdown of PLBD1-AS1 in tumor cells prevented the upregulation of PLBD1-AS1 in co-cultured PSCs (S2B Fig), and thus confirmed the exosomal transfer of PLBD1-AS1 from tumor cells to PSCs. CAFs promoted the proliferation of tumor cells (S2C Fig).We then evaluated markers of PSC activation to CAFs, including α-SMA, VIM, and FAP [36]. The results demonstrated that tumor-derived exosomes promote PSC activation into CAFs (Fig 4B, S2D Fig). We also measured IL-6, fibronectin, and MCP-1 [37] in the supernatant, indicating that exosome-induced CAF activation confers a pro-tumor phenotype (Fig 4C). Following the knockdown of PLBD1-AS1 expression in tumor cells, the exosomes exhibited a reduced ability to activate PSCs (Fig 4D, S2E and S2F Fig). Subsequently, we verified that the incubation of PSCs with tumor-derived exosomes significantly enhanced tumor cell proliferation (S2G Fig).

Furthermore, we demonstrated that tumor exosomes enhance glycolysis in PSCs via PLBD1-AS1, as evidenced by measurements of OCR and ECAR (Fig 4E and S2H Fig). It was further supported by increased lactate production, aldolase activity, and F1, 6 BP levels (Figs 4F and 4G, S2I Fig). To demonstrate whether the enhanced glycolysis promoted the activation of PSCs towards CAFs, we treated PSCs with low -glucose medium, 2-DG, or siALDOA. These treatments suppressed the tumor exosome-induced PSCs into CAFs (Figs 4H and 4I, S2J Fig). In summary, our findings indicate that tumor-derived exosomes boost glycolysis in PSCs via PLBD1-AS1, thereby promoting their activation into CAFs.

## Preparation and targeted validation of iRGD-exo-siPLBD1-AS1

The iRGD peptide specifically targets αvβ3 integrin and NRP-1 receptors on cell surfaces, facilitating vascular extravasation and tumor-specific penetration. To determine whether PDAC cells and PSCs are potential targets for iRGD, we evaluated the mRNA expression levels of αvβ3 integrin and NRP-1 in PDAC cells, PSCs, and HPNE. As expected, qPCR results (Fig 5A) revealed significantly elevated expression of αvβ3 integrin and NRP-1 in PDAC cells and PSCs compared to HPNE.

Subsequently, we isolated exosomes from the supernatant of mesenchymal stem cells (MSC) using ultracentrifugation. The exosomes were characterized through transmission electron microscopy (TEM) (Fig 5B) and nano-particle size analyzer (Fig 5C), confirming an average particle size of 96 nm, consistent with previous reported exosomal dimensions. Through Triton processing, we confirmed that the purity of the exosomes is approximately 50% (Fig 5D). Furthermore,

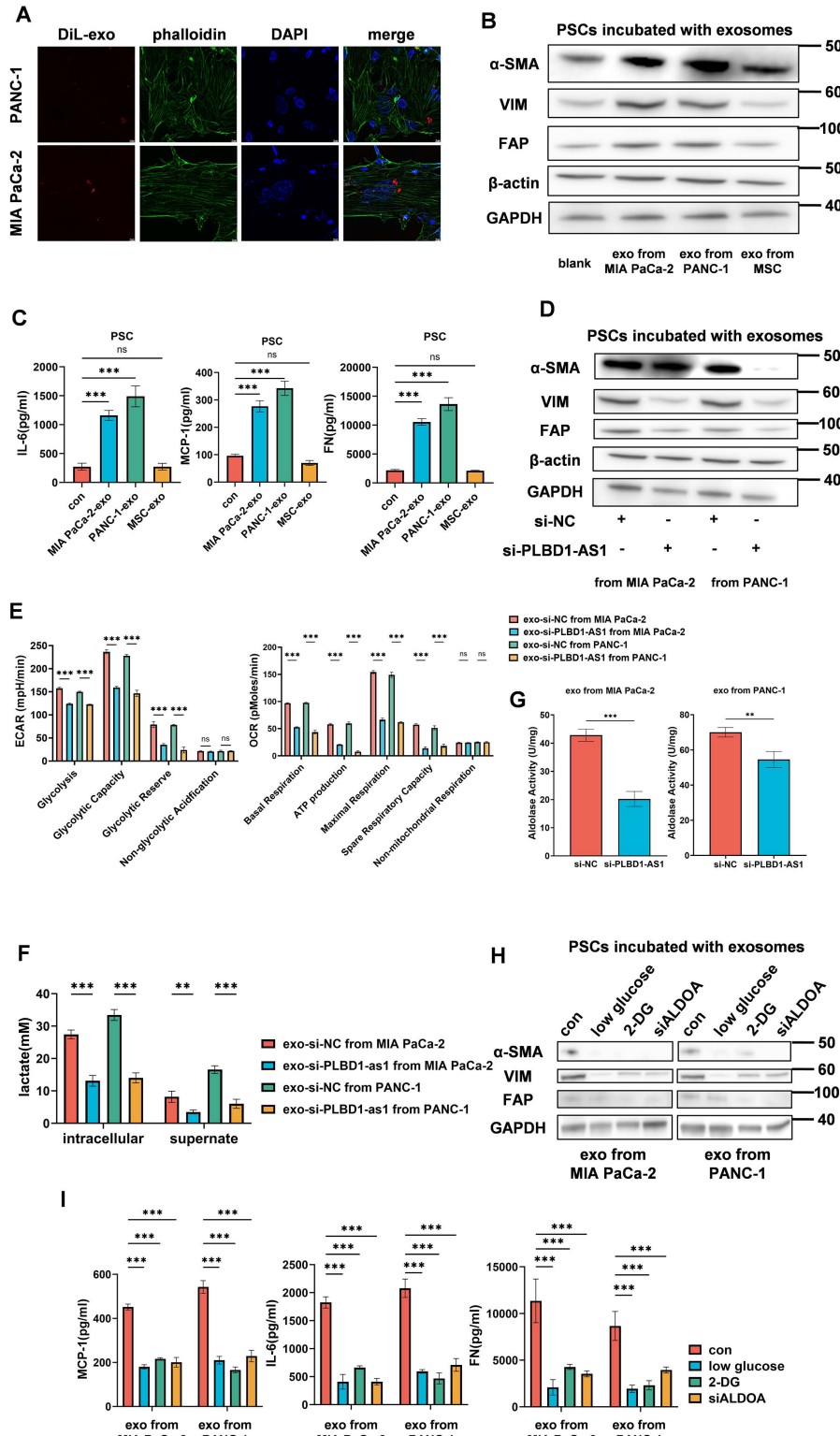

**Fig 4. PLBD1-AS1-exo promotes PSCs activation towards CAFs. (A)** Uptake of tumor cell-derived exosomes by PSCs. Exosomes from MIA PaCa-2 and PANC-1 cells were labeled with DiL (red), cytoskeleton with FITC-phalloidin (green), and nuclei with DAPI (blue) (n = 3). **(B and D)** Western blot

analysis of α-SMA, VIM, FAP, and β-actin in PSCs after treatment with tumor cell-derived exosomes (n = 3). **(C)** ELISA of IL-6, MCP-1, and FN levels in the supernatant of PSCs incubated with tumor cell-derived exosomes (n = 3). **(E)** ECAR and OCR measurements of PSCs following exosome treatment (n = 3). **(F)** Intracellular and extracellular lactate levels in PSCs after exosome incubation (n = 3). **(G)** ALDOA enzyme activity in PSCs treated with tumor cell-derived exosomes (n = 3). **(H)** Western blot analysis of α-SMA, VIM, FAP, and GAPDH in PSCs incubated with exosomes under low glucose, 2-DG, or ALDOA knockdown conditions (n = 3). **(I)** ELISA of IL-6, MCP-1, and FN in the supernatant of PSCs treated with exosomes under low glucose, 2-DG, or ALDOA knockdown (n = 3). Statistical analyses were performed by Student's t test (E, G and F) and one-way ANOVA (C and **I**). Bar graphs represent mean ± SEM. ns, not significant. *p < 0.05, **p < 0.01, ***p < 0.001.

we detected positive expression of exosomal markers (CD9, CD63, and CD81), and confirmed the absence of negative markers (calnexin and GAPDH) (Fig 5E), which collectively confirmed the exosomal identity. Using a lipid-peptide targeting kit, we anchored iRGD onto the exosomal membrane, which yielded iRGD-exo-siPLBD1-AS1 that exhibited a more pronounced efficacy in reducing expression levels in tumor cells (Fig 5F).

For the *in vitro* uptake study, we investigated whether PDAC cells and PSCs exhibit a higher affinity for iRGD-exos compared to unmodified exosomes. Cells were stained with DAPI (nuclei, blue), and DiL (membrane, red). Exosomes were labeled with DiO (green) and co-cultured with PDAC cells for 1 hour. Confocal imaging revealed markedly higher internalization of iRGD-exos than unmodified exosomes (Fig 5G), indicating that the iRGD modification significantly enhances cellular uptake. Flow cytometry analysis of cellular fluorescence after incubation with labeled exosomes (Fig 5H) confirmed enhanced uptake of iRGD-modified exosomes by both PSCs and MIA PaCa-2 cells.

To verify these findings *in vivo*, we administered DiD-labeled exosomes via tail vein injection to nude mice bearing PDAC xenografts. *In vivo* imaging at 3 hours post-injection showed stronger fluorescence signals in tumors from the iRGD-exo group compared to the control exosome group (Fig 5I). To visualize the distribution, organs were harvested and imaged. Images demonstrated that iRGD-exo accumulated predominantly in tumor tissue, with some uptake also observed in the liver. Importantly, tumor-specific accumulation of iRGD-exo was significantly higher than that of unmodified exosomes. These findings collectively demonstrate that iRGD-exo exhibits superior targeting and uptake efficiency both *in vitro* and *in vivo*. To evaluate the systemic and immunomodulatory safety profile of iRGD-exo-siPLBD1-AS1, we monitored body weight changes and measured serum levels of the pro-inflammatory cytokines IL-6 and TNF-α following treatment. No significant body weight loss was observed in mice treated with iRGD-exo-siPLBD1-AS1 compared to control groups, indicating a lack of overt systemic toxicity. Furthermore, serum levels of IL-6 and TNF-α were not significantly elevated, suggesting that the treatment did not induce a detectable systemic inflammatory response under the conditions tested. Since we observed accumulation of iRGD-exo in the liver and spleen, we performed H&E staining of these organs (Fig 5K) and measured serum AST levels (Fig 5L), which confirmed the absence of pathological changes in the liver and spleen. These data provide preliminary evidence for the safety of the engineered exosome formulation.

## In vitro and in vivo antitumor effect of iRGD-Exo-siPLBD1-AS1

To assess the impact of iRGD-exo-siPLBD1-AS1, we co-cultured iRGD-exo-siPLBD1-AS1 with tumor cells and observed a significant reduction in the levels of IL-6 and MCP-1 (Fig 6A) in the PSCs supernatant, indicating that iRGD-exo-siPLBD1-AS1 suppresses PSCs activation to CAFs.

We next evaluated the effect of engineered exosomes on PDAC cell proliferation using CCK-8 viability (Fig 6B) assays and EdU assays (S3A Fig) after co-culture with PDAC cells. Exosomes loaded with siPLBD1-AS1 effectively inhibited the proliferation of PDAC cells, enhanced by the incorporation of iRGD. Consistent with these findings, colony formation assay (S3B Fig) showed that iRGD-exo-siPLBD1-AS1 markedly reduced the clonogenic capacity of PDAC cells.

To further explore the influence of exosomes on PDAC cells, we performed wound healing (S3C Fig) and transwell invasion assays (Fig 6C). iRGD-Exo-siPLBD1-AS1 significantly impaired both migratory and invasive abilities of PDAC cells. These data suggest that exosomes can effectively deliver siPLBD1-AS1 into target cells and exert biological effects, with iRGD further improving cellular uptake and enhancing their anticancer activity.

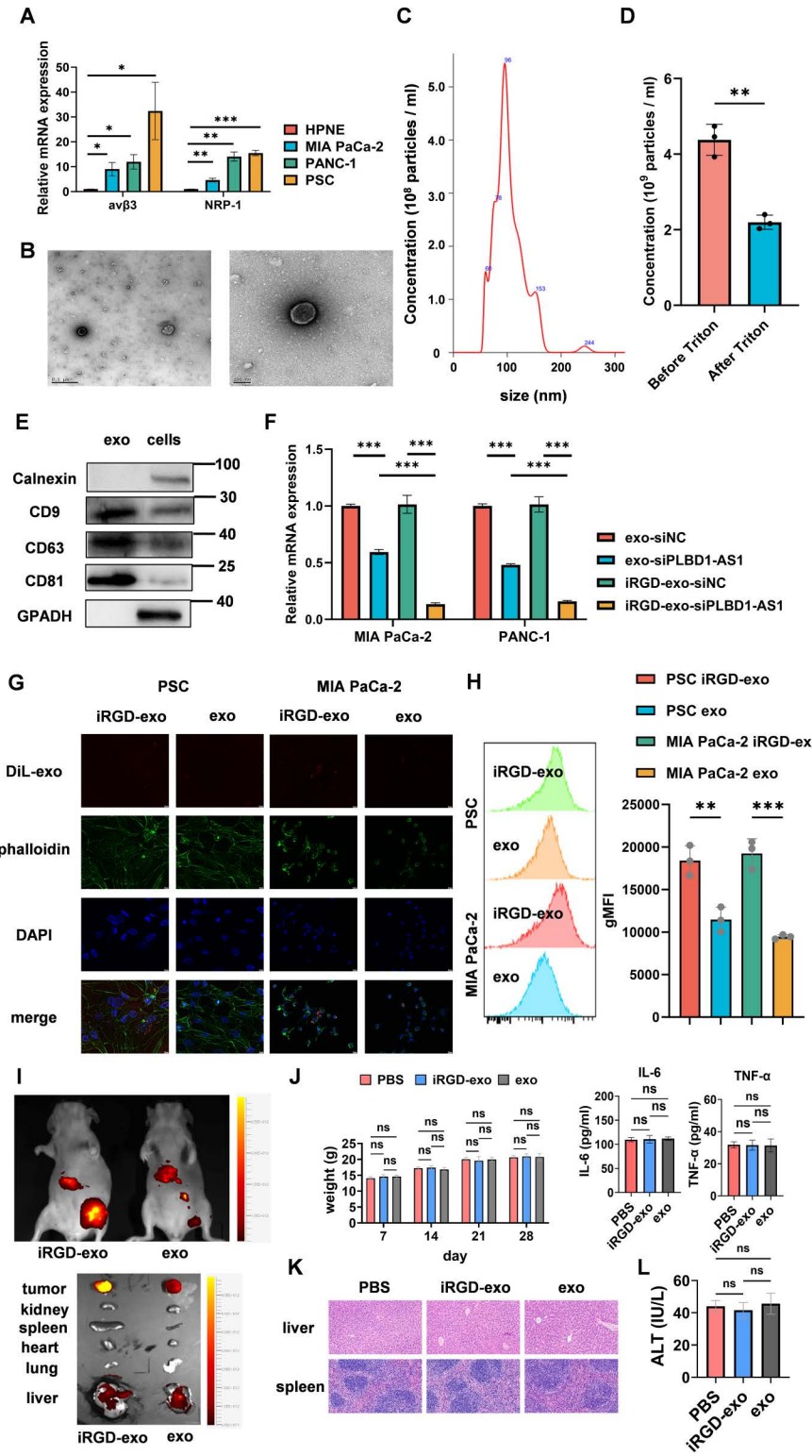

**Fig 5. Preparation and validation of targeted iRGD-exo-siPLBD1-AS1. (A)** Relative mRNA expression of integrins αvβ3 and NRP-1 in HPNE, MIA PaCa-2, PANC-1, and PSC cell lines (n = 3). **(B)** TEM images of exosomes isolated from MSCs. **(C)** (left) Size distribution of exosomes as determined by nanoparticle tracking analysis. **(D)** Quantitative analysis of exosomal particles before and after Triton treatment (n = 3). **(E)** Western blot

analysis for positive exosomal markers (CD9, CD63, CD81) and negative markers (calnexin, GAPDH) in exosomes and MSCs (n = 3). **(F)** Relative PLBD1-AS1 mRNA levels in MIA PaCa-2 and PANC-1 cells after treatment with engineered exosomes (n = 3). **(G)** Cellular uptake of engineered exosomes in PSCs and MIA PaCa-2 cells. Exosomes (DiL, red), cytoskeleton (FITC-phalloidin, green), and nuclei (DAPI, blue) are shown (n = 3). **(H)** Representative flow cytometry images (left) and quantification of mean fluorescence intensity (MFI, right) showing the uptake of fluorescently labeled exosomes by PSCs and MIA PaCa-2 cells (n = 3). **(I)** *In vivo* distribution of DiL-labeled exosomes (red) following intravenous injection in a mouse xenograft model, showing fluorescence in tumors and major organs (n = 5). **(J)** Measurements of body weight and serum inflammatory cytokines (IL-6 and TNF-α) after PBS or exosome treatment (n = 5). **(K)** Histopathological evaluation of liver and spleen tissues. H&E staining shows normal tissue architecture in both iRGD-exo-treated and control groups, indicating absence of treatment-related pathological changes (n = 5). **(L)** Assessment of hepatic function following iRGD-exo administration. Serum aspartate aminotransferase (AST) levels remain within normal range, confirming the absence of hepatotoxicity (n = 5). Statistical analyses were performed by Student's t test (D and H) and one-way ANOVA (A, F, J and **L**). Bar graphs represent mean ± SEM. ns, not significant. *$p < 0.05$, **$p < 0.01$, ***$p < 0.001$.

To evaluate the therapeutic potential of iRGD-exo-siPLBD1-AS1 *in vivo*, we established MIA PaCa-2 xenografts tumors in nude mice. After two weeks of tumor growth, animals received 100 µg of exosomes/iRGD-exos every 2 days for another 2 weeks (Fig 6D). Treatment with iRGD-exo-siPLBD1-AS1 significantly suppressed tumor growth *in vivo* (Figs 6E-6G). In addition, the lactate levels within tumors were significantly lower in the treatment group than in controls (S3D Fig). The Ki-67 staining results on tumor tissue indicated that iRGD-exo-siPLBD1-AS1 significantly inhibited tumor cell proliferation (Fig 6H). Immunostaining for αSMA further demonstrated that exosome treatment effectively reduced the activation of PSCs into CAFs (Fig 6I).

## Discussion

This study investigated the oncogenic effect of a novel lncRNA, PLBD1-AS1, in PDAC. We found that PLBD1-AS1 is highly expressed in tumor cells, promoting proliferation, migration, and invasion. Additionally, PLBD1-AS1 can be secreted via exosomes and subsequently internalized by PSCs, driving their activation into CAFs and thereby accelerating tumor progression. Based on these findings, we developed engineered exosomes as delivery vehicles for siPLBD1-AS1, successfully demonstrating the efficacy of RNAi-based therapy both *in vitro* and *in vivo*.

The progression of tumors facilitated by PLBD1-AS1 involves multiple mechanisms. It interacts directly with ALDOA, a key glycolytic enzyme frequently overexpressed in pan-tumors and linked to poor prognosis. We demonstrated that PLBD1-AS1 enhances ALDOA enzymatic activity, leading to increased glycolytic flux and elevated levels of metabolic intermediates such as lactate. Given that lactate is known to promote tumor cell proliferation, immune evasion, and metastasis. Silencing PLBD1-AS1 significantly reduced lactate production and impaired glycolysis. Glycolysis serves as a crucial metabolic pathway for the proliferation, migration, and metastasis of tumor cells. Mechanistically, PLBD1-AS1 knockdown strengthened the interaction between ALDOA and ATP6V1B2, facilitating the activation of AMPK and subsequent suppression of biosynthetic pathways, thereby attenuating tumorigenicity.

Furthermore, we investigated the transcellular role of PLBD1-AS1 via exosomes. PSCs efficiently internalized tumor-derived exosomes, leading to their activation into CAFs, which secreted IL-6, MCP-1, and fibronectin to remodel the tumor stroma and support malignant progression. This activation was dependent on exosomal PLBD1-AS1, as its silencing in tumor cells abolished PSCs activation. Similarly, PLBD1-AS1-exo increased the glycolysis level of PSCs. Following glucose deprivation or glycolytic inhibition, the activation of tumor-derived exosomes on PSCs was prevented, underscoring the necessity of glycolytic reprogramming in this activation. Thus, targeting PLBD1-AS1 disrupts a key intercellular signaling axis that sustains tumor-stroma crosstalk.

In addition to its identified role in post-translationally regulating ALDOA activity, we have also considered the potential cis-regulatory function of PLBD1-AS1 on its sense gene, PLBD1. Although PLBD1 has been reported to be upregulated in several tumor types [38,39], its functional impact on tumor progression remains unclear. Our current data indicate that the oncogenic effects of PLBD1-AS1 are primarily mediated through its direct interaction with ALDOA and the subsequent

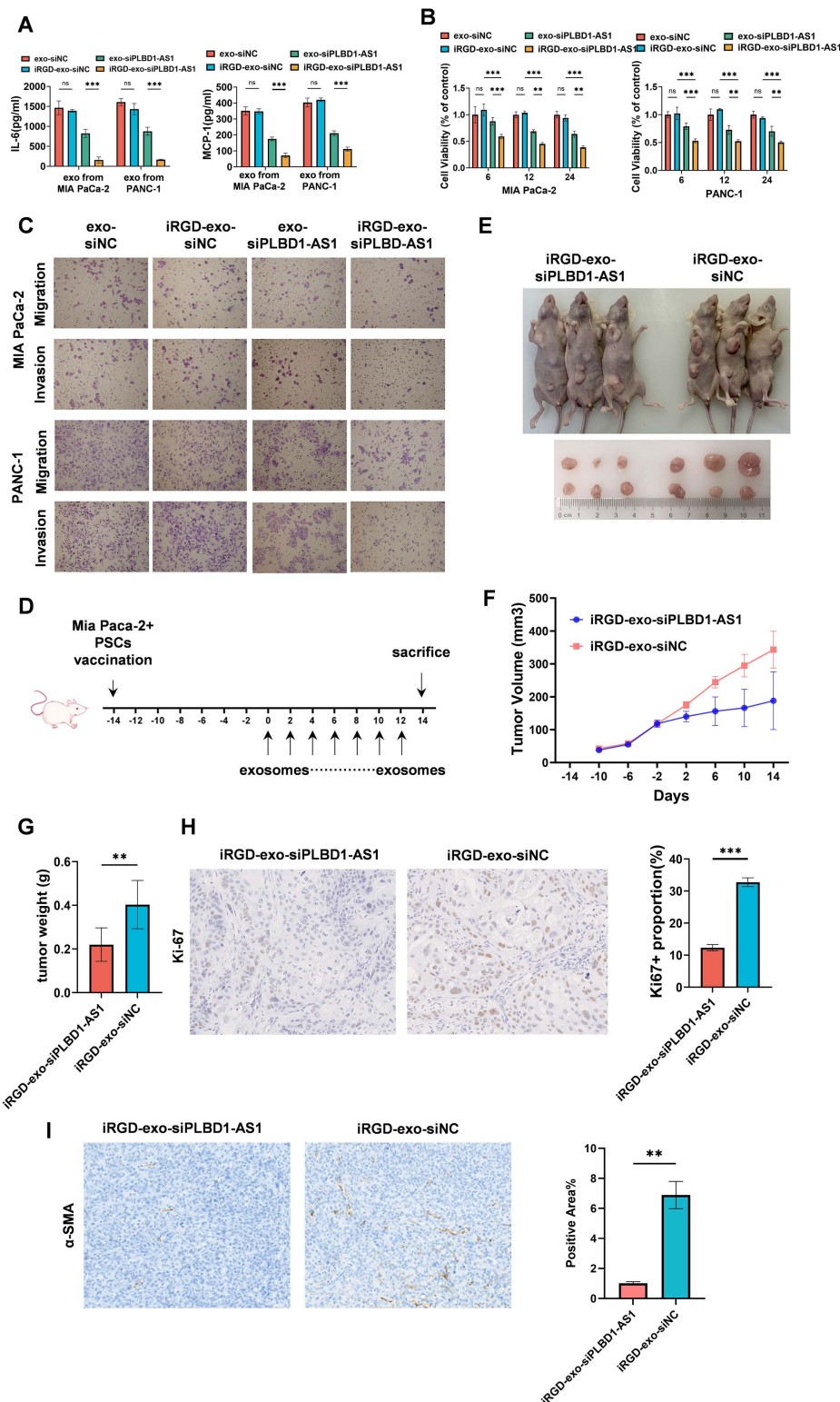

**Fig 6. Antitumor efficacy of iRGD-Exo-siPLBD1-AS1 *in vitro* and *in vivo*. (A)** ELISA analysis of IL-6 and MCP-1 concentrations in the supernatant of PSCs treated with engineered exosomes (n = 3). **(B)** Viability of MIA PaCa-2 and PANC-1 cells treated with engineered exosomes, assessed by CCK-8 assay (n = 3). **(C)** Representative images of transwell migration assays for MIA PaCa-2 and PANC-1 following incubation with engineered exosomes. **(D)**

Schematic of the subcutaneous xenograft model. Tumors were established in BALB/c nude mice by co-inoculating MIA PaCa-2 cells and PSCs. Mice received bi-daily exosome treatments for two weeks. **(E-G)** Gross morphology, weights, and volumes of subcutaneous tumors (n = 3). **(H)** Representative image and quantitative analysis of Ki67 and αSMA immunohistochemical staining in tumor tissue (n = 3). **(I)** Representative image and quantitative analysis of αSMA immunohistochemical staining in tumor tissue (n = 3). Statistical analyses were performed by Student's t test (A, B, G, H and **I**). Bar graphs represent mean ± SEM. ns, not significant. *$p < 0.05$, **$p < 0.01$, ***$p < 0.001$.

enhancement of glycolytic flux. Nevertheless, we cannot exclude a potential contribution from modulated PLBD1 expression. Future studies will be valuable to delineate whether and how the regulation of PLBD1 by PLBD1-AS1 might cooperate with the ALDOA–glycolysis axis to drive PDAC progression.

Given the unique oncogenic role of PLBD1-AS1, we developed a targeted RNAi strategy using exosomes functionalized with the iRGD peptide, which selectively binds integrin αvβ3 and NRP-1 to enhance tumor and stromal targeting. While traditional therapeutics focus on inhibiting protein products, their efficacy is often constrained by the challenging physicochemical properties of proteins and their complex interactomes [40–42]. In contrast, RNA-based therapeutics, especially siRNA, emerge as a transformative strategy by targeting disease at the transcriptional level, thereby avoiding the pitfalls of protein-targeting [43]. In this study, we utilized exosomes as an optimal delivery platform for siRNA, ensuring its stability *in vivo*. By engineering these exosomes with the iRGD peptide to target tumors and PSCs, we achieved efficient delivery of siPLBD1-AS1 and effectively suppressed its oncogenic activity, demonstrating a viable RNAi therapy for pancreatic cancer.

While our subcutaneous xenograft model provided valuable initial insights into the in vivo efficacy of iRGD-exo-siPLBD1-AS1, we acknowledge that this model does not fully recapitulate the complex tumor microenvironment (TME) and metastatic propensity of human PDAC. The subcutaneous location lacks the organ-specific stroma, vascular network, and immune context of the pancreas. Future studies employing orthotopic pancreatic models or patient-derived xenografts (PDX) would be highly valuable. An orthotopic model would allow for the assessment of tumor growth and therapy response within the appropriate anatomical and microenvironmental context, while PDX models, which retain the stromal and genetic heterogeneity of human tumors, could provide even more clinically relevant data on the therapeutic potential and stromal remodeling capabilities of our engineered exosomes.

In summary, iRGD-Exo-siPLBD1-AS1 represents a potent therapeutic strategy for PDAC, with strong clinical translation potential. Our work not only elucidates the oncogenic role of PLBD1-AS1 in PDAC, but also establishes a foundational basis for innovative therapies aimed at stromal reprogramming and tumor inhibition.

## Conclusions

Our study identifies the exosomal lncRNA PLBD1-AS1 as a key driver of pancreatic cancer. PLBD1-AS1 boosts tumor cell malignancy by activating ALDOA and reprograms PSCs into CAFs. To target this pathway, we engineered iRGD-exosomes to deliver siPLBD1-AS1, which effectively inhibited tumor growth and stromal activation. This work highlights the potential of exosome-mediated RNAi as a novel therapy for pancreatic cancer.

## Supporting information

**S1 Fig.** (A) Survival analysis of ALDOA in the TCGA cohort. Quantitative analysis of Western blot for ALDOA and β-actin in MIA PaCa-2 and PANC-1 cell lines (n = 3). Extracellular acidification rate (ECAR) and oxygen consumption rate (OCR) in MIA PaCa-2 and PANC-1 cells. Quantitative analysis of AMPK pathway protein phosphorylation following PLBD1-AS1 knockdown (n = 3). Quantitative analysis of Western blot of phosphorylated AMPK pathway proteins in PANC-1 with single or dual knockdown of PLBD1-AS1 and ATP6V1B2 (n = 3). Statistical analyses were performed by Student's t test (A and D) and one-way ANOVA (E). Bar graphs represent mean ± SEM. ns, not significant. *$p < 0.05$, **$p < 0.01$, ***$p < 0.001$. (PDF)

**S2 Fig.** (A, B) Relative PLBD1-AS1 mRNA expression in PSCs after exosome treatment (n = 3).(C) Viability of MIA PaCa-2 and Panc1 cells co-cultured with PSCs or CAFs, assessed by CCK-8 assay (n = 3). (D, E) Quantitative analysis of α-SMA, VIM, FAP, and β-actin protein expression in PSCs following treatment with tumor cell-derived exosomes, as determined by Western blot (n = 3). (F) Secreted levels of IL-6, MCP-1, and FN in MIA PaCa-2 and Panc-1 cell supernatants, measured by ELISA (n = 3). (G) Representative images and quantitation of EdU assay showing proliferation of MIA PaCa-2 and PANC-1 cells co-cultured with PSCs 0or CAFs. (H) Extracellular acidification rate (ECAR) and oxygen consumption rate (OCR) of PSCs following treatment with MIA PaCa-2- or PANC-1-derived exosomes. (I) Fructose-1,6-bisphosphate (F1,6 BP) levels in PSCs after treatment with tumor cell-derived exosomes (n = 3). (J) Quantitative analysis of α-SMA, VIM, FAP, and GAPDH in PSCs incubated with exosomes under low glucose, 2-DG, or ALDOA knockdown conditions (n = 3). Statistical analyses were performed by Student's t test (B, C, E, F, I and J) and one-way ANOVA (A and D). Bar graphs represent mean ± SEM. ns, not significant. *p < 0.05, **p < 0.01, ***p < 0.001.
(PDF)

**S3 Fig.** (A-C) Proliferation, clonogenicity, and migration of MIA PaCa-2 and PANC-1 cells treated with the indicated exosomes, as assessed by (A) EdU, (B) colony formation, and (C) wound healing assays (n = 3). (D) Lactate concentration in tumor tissues from nude mice treated with iRGD-exo-siNC or iRGD-exo-siPLBD1-AS1 (n = 3). Statistical analyses were performed by Student's t test (B and D) and one-way ANOVA (C). Bar graphs represent mean ± SEM. ns, not significant. *p < 0.05, **p < 0.01, ***p < 0.001.
(PDF)

## Author contributions

**Conceptualization:** Wenbo Zhu, Lei Li.

**Data curation:** Xianzhu Zhou.

**Formal analysis:** Jiayu Chen.

**Funding acquisition:** Lei Li.

**Investigation:** Yating Zhao.

**Methodology:** Weina Hao, Xianzhu Zhou.

**Project administration:** Yiqi Du.

**Resources:** Xianzhu Zhou.

**Software:** Congjia Ma.

**Supervision:** Xiangyu Kong.

**Validation:** Wenbo Zhu, Xintong Zhao.

**Visualization:** Xiangyu Kong.

**Writing – original draft:** Wenbo Zhu.

**Writing – review & editing:** Xiangyu Kong.

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
