## [Decision Letter · Decision Letter 0]

20 Oct 2025

Dear Dr. Li,

Thank you for submitting your manuscript to PLOS ONE. After careful consideration, we feel that it has merit but does not fully meet PLOS ONE’s publication criteria as it currently stands. Therefore, we invite you to submit a revised version of the manuscript that addresses the points raised during the review process.

We look forward to receiving your revised manuscript.

Kind regards,

Mohammad H. Ghazimoradi

Academic Editor

PLOS ONE

[Supported in part by grant 82172572 (LL), 82170659 (YQD), 82072760 and 82473057 (XYK) from the National Natural Science Foundation of China.].

Additional Editor Comments (if provided):

Reviewers' comments:

Reviewer's Responses to Questions

**Comments to the Author**

1. Is the manuscript technically sound, and do the data support the conclusions?

Reviewer #1: Partly

Reviewer #2: Yes

Reviewer #3: Yes

2. Has the statistical analysis been performed appropriately and rigorously?

Reviewer #1: No

Reviewer #2: Yes

Reviewer #3: Yes

3. Have the authors made all data underlying the findings in their manuscript fully available?

Reviewer #1: Yes

Reviewer #2: No

Reviewer #3: Yes

4. Is the manuscript presented in an intelligible fashion and written in standard English?

Reviewer #1: No

Reviewer #2: Yes

Reviewer #3: Yes

Reviewer #1: The manuscript addresses a potentially interesting topic and presents results that could be promising. However, in its current form, the work does not meet the standards required for publication. Major issues concern both the clarity of presentation and methodological aspects. Below I outline the main weaknesses.

General comments:

Abbreviations: these are not consistently defined or used. Some are never introduced, others are written in different ways throughout the text (e.g., variations in capitalization, use of hyphens), and some are used only once, which makes them unnecessary.

Figures: the numbering of figures in the Results section is incorrect and needs to be corrected.

Language and typos: several typos are present throughout the manuscript and should be corrected.

Below I provide more detailed comments organized by section.

Abstract:

The abstract is written in unclear English and would benefit from language editing. In addition, there are several typos(e.g., line 28: oncogenic is incorrectly written as on-cogenic). Abbreviations are also not defined when first introduced, which makes the text difficult to follow.

Introduction:

The introduction is generally clear and appropriate. However, there is a typo at line 86, where αvβ5 is reported instead of αvβ3. This should be corrected.

Materials and Methods:

This section requires substantial revision. In its current form, it includes some unnecessary details while omitting essential information needed for reproducibility.

Major methodological concerns:

Line 133: the use of 10% FBS is reported. Was this FBS exosome-depleted? Otherwise, results could be confounded by exosomes present in serum.

Section 2.7: The method described does not allow isolation of exosomes but rather larger microvesicles. Isolation of exosomes requires ultracentrifugation at 100,000 g.

Line 231: lipofectamine 2000 is used. What was the transfection efficiency? Wouldn’t RNAiMAX, specifically designed for siRNA, be more appropriate?

Minor concerns:

Databases: only a few examples are reported. All databases used should be listed, ideally in a table, to ensure reproducibility.

Line 102: it is not clear whether the reported p-values for DEGs are adjusted. Later in the Results section FDR is mentioned—this needs clarification.

Lines 112–113: the described method is unclear. Please specify.

Lines 141–142: a reference for the cited method is missing.

Line 149–152: ELISA for FN1 is mentioned at line 149 but not referred to in lines 150–152. Please clarify this inconsistency.

Lines 204–205: the described method is unclear. Please specify.

Line 208: mass spectrometry is mentioned, but no methodological details are provided.

Paragraph 2.19: the description is unclear and should be rewritten for clarity.

Line 283: normalization by cell number is reported. How was the cell number determined?

Results

The results are generally well reported. However, there are some important omissions:

Section 3.3: The cell type used for the experiments is not specified.

Lines 395–396 and 399–401: figures and statistical analyses are missing, which prevents proper interpretation of the results.

Discussion

The discussion is overall clear and appropriate in its current form.

Figures

Overall, the figures are adequate. However, there are some issues that need to be addressed:

Figure 3A: The meaning of the two circles is not clear and should be clarified.

Fig. S5B: The resolution is too low, making it impossible to assess whether the cells have migrated or not. Higher quality images are required.

Reviewer #2: The manuscript presents a novel and well-designed study on the oncogenic role of PLBD-as1 in pancreatic ductal adenocarcinoma (PDAC) and proposes iRGD-modified exosomes as an efficient siRNA delivery vehicle. The integration of bioinformatics, mechanistic assays, and in vivo models makes the work important and timely. Overall, the manuscript has strong translational potential, and I recommend publication after revision.

Major Points

• Exosome characterization: Please expand quantitative data (particle concentration, purity index) to strengthen Fig.5 (lines ~423–429).

• In vivo validation: Current experiments use only subcutaneous xenografts (lines ~209–219, 471–478). Consider acknowledging this limitation and discussing the value of orthotopic or PDX models.

• Mechanistic evidence: The PLBD-as1–ALDOA interaction (lines ~353–382) would benefit from rescue experiments or additional discussion of potential follow-up studies.

• Toxicity and biodistribution: Since exosomes also accumulate in liver/spleen (lines ~441–447), please provide toxicity assessment or state that this will be evaluated in future studies.

• Clinical samples: Lines ~315–341 describe cell-line validation; inclusion or at least discussion of PDAC patient tissue analysis would enhance translational relevance.

• Data availability: RNA-seq/exosomal sequencing datasets (lines ~91–103, 315–325) should be deposited in a public repository for transparency.

Minor Points

• Line 27: “on-cogenic” → “oncogenic.”

• Lines 27–29, 49–89: Ensure consistency in naming PLBD-as1 vs. PLBD-AS1.

• Lines 135–147: Clarify centrifugation speeds during exosome isolation (10,000 g vs. 12,000 g).

• Line 371: Add clarification on normalization method in Seahorse assay (OCR/ECAR).

• References (line ~536 onwards): Please update with recent exosome therapy literature (2020–2024).

Overall Recommendation

This is a strong and innovative manuscript with potential clinical relevance. With the above revisions and clarifications, it will make a valuable contribution to the field of PDAC therapeutics. I recommend Major Revision with a view to acceptance after revision.

Reviewer #3: Zhu et al explored the mechanism of tumor progression mediated by tumor exosomes. The authors did a comprehensive study including identifying the potential oncogenic lncRNA from exosomes, explaining the potential mechanism and engineering exosomes for RNAi delivery as a promising intervention strategy for pancreatic cancer. I think the paper is suitable for publication, but needs further modification:

1.I think the authors meant to describe about PLBD1-as1 throughout the paper. But most of the time it was typed as “PLBD-as1”. This needs to be paid attention and corrected before publication.

2.The source of the exosome sequencing data (from five healthy individuals and five patients) isn’t very clear to me. If it is from the database, please describe further about the screening strategy for the patients. If it is collected and processed by the authors, they should also describe about the specimen collecting methods.

3.In method section 2.7(line138), I believe the 10000 and 12000g centrifugation is a typo. Exosomes may not pellet at this speed.

4.In method 2.8(line 141), please cite Ding’s method if it is not described in detail.

5.In result section 3.4 (line384-413), all figure numbers are mistakenly labeled. Please correct it.

6.For figure 5e, it shows that iRGD-exo-siPLBD1-as1 has better efficacy than exo-siPLBD1-as1. Is there a control set with exo-siNC to show the fold change?

7.For figure 5f, the internalization efficiency change isn’t very obvious shown by the image, especially for the PSC set. Is it better to analyze using flow method to show the quantitative data?

8.The entire paper is discussing about the PLBD1-as1 binder. But the main function for PLBD1-as1 is to alter the PLBD1 expression, would be nice to have some discussion about that.

.

Reviewer #1: No

Reviewer #2: **Yes:**DIBYASHREE CHHETRIDIBYASHREE CHHETRIDIBYASHREE CHHETRIDIBYASHREE CHHETRI

Reviewer #3: No

---

## [Author Response · Author response to Decision Letter 1]

5 Dec 2025

Dear Editors,

Thank you very much for your letter and for the opportunity to submit a revised version of our manuscript entitled “Engineering Exosomes with iRGD for Targeted RNAi Therapy against Pancreatic Cancer Mediated by Long Non-coding RNA PLBD1-AS1” (ID: PONE-D-25-31245). We sincerely appreciate the time and effort you and the reviewers have dedicated to evaluating our work.

We are grateful to the reviewers for their thoughtful and constructive comments, which have helped us significantly improve the quality and clarity of our manuscript. We have carefully addressed each of the points raised and have revised the manuscript accordingly.

In this revision, we have undertaken thorough language editing and formatting refinements to enhance the overall readability and adherence to journal guidelines. The key revisions include: standardized use of abbreviations, clarification of methodological details, additional quantitative data and controls, expanded discussion of study limitations, and enhanced figure quality. All changes have been implemented with careful attention to the reviewers' and editor's suggestions.

We believe the manuscript has been substantially strengthene. Please find attached our point-by-point response to reviewers, a marked-up manuscript highlighting all revisions, and the raw manuscript.

Thank you again for your time and consideration.

Sincerely,

Lei Li,

---

## [Decision Letter · Decision Letter 1]

22 Dec 2025

Dear Dr. Li,

Thank you for submitting your manuscript to PLOS ONE. After careful consideration, we feel that it has merit but does not fully meet PLOS ONE’s publication criteria as it currently stands. Therefore, we invite you to submit a revised version of the manuscript that addresses the points raised during the review process.

We look forward to receiving your revised manuscript.

Kind regards,

Mohammad H. Ghazimoradi

Academic Editor

PLOS One

Journal Requirements:

Reviewers' comments:

Reviewer's Responses to Questions

**Comments to the Author**

Reviewer #2: All comments have been addressed

Reviewer #3: All comments have been addressed

2. Is the manuscript technically sound, and do the data support the conclusions?

Reviewer #2: Yes

Reviewer #3: Yes

3. Has the statistical analysis been performed appropriately and rigorously?

Reviewer #2: No

Reviewer #3: Yes

4. Have the authors made all data underlying the findings in their manuscript fully available?

Reviewer #2: No

Reviewer #3: Yes

5. Is the manuscript presented in an intelligible fashion and written in standard English?

Reviewer #2: Yes

Reviewer #3: Yes

Reviewer #2: Major Points

1. The exosomal RNA-seq discovery cohort (5 PDAC vs 5 controls) is small; please clarify whether multiple-testing correction was applied and justify selection of PLBD1-AS1.

2. The prognostic model is developed and tested in the same TCGA cohort; independent validation and multivariate Cox analysis including clinical variables are recommended.

3. The interaction between PLBD1-AS1 and ALDOA is supported by RNA pull-down but would benefit from additional validation (e.g., RIP-qPCR or rescue experiments).

4. The ALDOA–vATPase–AMPK pathway is not fully characterized; assessment of downstream AMPK targets and functional dependency would strengthen the mechanistic claims.

5. PSC activation into CAFs is convincingly shown in vitro but lacks in vivo confirmation using CAF markers (e.g., α-SMA, FAP, collagen deposition).

6. The use of subcutaneous xenograft models limits translational relevance; a brief discussion of this limitation and justification of exosome dosing is suggested.

7. Safety evaluation is limited to H&E staining and AST levels; additional discussion or assessment of systemic and immunological safety would strengthen the therapeutic claims.

Minor Points

8. Please standardize nomenclature and use 'PLBD1-AS1' consistently throughout the manuscript.

9. Minor grammatical and stylistic issues remain; careful language editing is recommended.

10. The term “oncogene” may be replaced with “oncogenic lncRNA” to better reflect the functional evidence.

11. Details regarding biological replicates, randomization, and blinding should be clearly stated.

12. Figures, particularly Western blots, would benefit from molecular weight markers and clearer quantification information.

Reviewer #3: The authors well explained the potential mechanism and engineered exosomes for RNAi delivery as a promising intervention strategy for pancreatic cancer. They solved my questions well. I have no more comments and think it is suitable for publication.

.

Reviewer #2: **Yes:**DIBYASHREE CHHETRIDIBYASHREE CHHETRIDIBYASHREE CHHETRIDIBYASHREE CHHETRI

Reviewer #3: No

---

## [Author Response · Author response to Decision Letter 2]

7 Feb 2026

Dear Editor and Reviewers,

Thank you once again for your valuable feedback and for giving us the opportunity to revise our manuscript. We have carefully addressed all the comments and suggestions raised during the second round of review. The revised manuscript has been updated accordingly, and we have also uploaded the point-by-point response file detailing all changes made.

We believe the manuscript has been significantly improved through this revision process. Please let us know if any further information is required.

---

## [Decision Letter · Decision Letter 2]

9 Mar 2026

Engineering Exosomes with iRGD for Targeted RNAi Therapy against pancreatic cancer mediated by long non-coding RNA PLBD1-AS1

PONE-D-25-31245R2

Dear Dr. Li,

We’re pleased to inform you that your manuscript has been judged scientifically suitable for publication and will be formally accepted for publication once it meets all outstanding technical requirements.

Kind regards,

Manasa Varra

Academic Editor

PLOS One

Additional Editor Comments (optional):

Dear Wenbo Zhu,

we are happy to inform that the revised manuscript PONE-D-25-31245R2 entitled "Engineering Exosomes with iRGD for Targeted RNAi Therapy against Pancreatic Cancer Mediated by long non-coding RNA PLBD1-AS1" is hereby recommended for acceptance for publication in PLOS One as all the comments raised have been addressed.

Reviewers' comments:

Reviewer's Responses to Questions

**Comments to the Author**

Reviewer #2: All comments have been addressed

Reviewer #3: All comments have been addressed

2. Is the manuscript technically sound, and do the data support the conclusions?

Reviewer #2: Yes

Reviewer #3: Yes

3. Has the statistical analysis been performed appropriately and rigorously?

Reviewer #2: Yes

Reviewer #3: Yes

4. Have the authors made all data underlying the findings in their manuscript fully available?

Reviewer #2: Yes

Reviewer #3: Yes

5. Is the manuscript presented in an intelligible fashion and written in standard English?

Reviewer #2: Yes

Reviewer #3: Yes

Reviewer #2: The manuscript is strong and carefully executed. I have only a few minor suggestions to improve clarity and reporting transparency:

Please briefly clarify normalization and batch correction methods for the GEO dataset, given the relatively small sample size.

Provide a short justification for the in vivo sample size (n = 3 per group), even if based on pilot data.

If available, include quantitative information on siRNA loading efficiency in engineered exosomes.

Confirm availability of raw proteomics and Seahorse data in accordance with journal data policies.

Standardize statistical software version reporting and minor formatting inconsistencies.

These points are minor and do not detract from the overall scientific validity of the study.

Reviewer #3: The authors have addressed all my previous questions. The revisions have significantly improved the manuscript, and I have no more comments.

.

Reviewer #2: No

Reviewer #3: No

---

## [Editor Report · Acceptance letter]

PONE-D-25-31245R2

PLOS One

Dear Dr. Li,

I'm pleased to inform you that your manuscript has been deemed suitable for publication in PLOS One. Congratulations! Your manuscript is now being handed over to our production team.

Kind regards,

on behalf of

Dr. Manasa Varra

Academic Editor

PLOS One